# Improved betulinic acid biosynthesis using synthetic yeast chromosome recombination and semi-automated rapid LC-MS screening

G.-O.F. Gowers [1,2], S.M. Chee[3,4], D. Bell [3,4,5], L. Suckling[3,4,5], M. Kern[6], D. Tew [6], D.W. McClymont[3,4] & T. Ellis [1,2 ✉]

Synthetic biology, genome engineering and directed evolution offer innumerable tools to expedite engineering of strains for optimising biosynthetic pathways. One of the most radical is SCRaMbLE, a system of inducible in vivo deletion and rearrangement of synthetic yeast chromosomes, diversifying the genotype of millions of *Saccharomyces cerevisiae* cells in hours. SCRaMbLE can yield strains with improved biosynthetic phenotypes but is limited by screening capabilities. To address this bottleneck, we combine automated sample preparation, an ultra-fast 84-second LC-MS method, and barcoded nanopore sequencing to rapidly isolate and characterise the best performing strains. Here, we use SCRaMbLE to optimise yeast strains engineered to produce the triterpenoid betulinic acid. Our semi-automated workflow screens 1,000 colonies, identifying and sequencing 12 strains with between 2- to 7-fold improvement in betulinic acid titre. The broad applicability of this workflow to rapidly isolate improved strains from a variant library makes this a valuable tool for biotechnology.

[1] Imperial College Centre for Synthetic Biology, Imperial College London, London SW7 2AZ, UK. [2] Department of Bioengineering, Imperial College London, London SW7 2AZ, UK. [3] London Biofoundry, Imperial College London, London SW7 2AZ, UK. [4] SynbiCITE, Imperial College London, London SW7 2AZ, UK. [5] Structural and Synthetic Biology, Department of Infectious Disease, Imperial College London, London SW7 2AZ, UK. [6] GlaxoSmithKline, Stevenage SG1 2NY, UK. ✉email: t.ellis@imperial.ac.uk

Modifying cells to improve production of a desired endogenous or heterologous metabolite is a broad aim for many areas of academic and industrial biotechnology and biosciences[1]. Classic approaches to metabolic engineering use tools such as knockout libraries and flux-balance analysis, among others, to predict rational changes to both the biosynthetic pathway and the native metabolism of the host organisms that will improve metabolite titre[2–4]. Synthetic biology now offers many tools to rapidly modify pathways and genomes to optimise metabolite titres. For example, modular DNA pathway construction methods are ubiquitous and now available for a variety of microorganisms[5–10]. The nascent field of synthetic genomics also now allows entirely new chromosome designs that introduce new tools for genome optimisation for metabolic engineering purposes. The synthetic yeast project (Sc2.0) aims to synthesise each yeast chromosome from chemical parts based on in silico design, with one of the features implemented in these designs being the introduction of 34 bp loxP-sym recombination sites downstream of every non-essential gene[11,12]. Upon chemical induction of a plasmid-expressed Cre recombinase, recombination takes place between chromosomal loxP-sym sites in the living cell nucleus, leading to gene deletion, duplication, and rearrangements[13,14]. The rapid creation of host genome diversity by SCRaMbLE has been demonstrated to improve both native and engineered phenotypes. In previous works, synthetic chromosomes have been SCRaMbLEd and yeast strains selected with improved tolerance to either high alkali conditions[15], caffeine[16], heat[16,17], ethanol[17], or acetic acid[17]. SCRaMbLE has also been used to improve the performance of yeast strains in biosynthesis of heterologous metabolites, including strains engineered with pathways for violacein[16,18,19] and for carotene and lycopene[16,20,21].

These past studies all improved phenotypes that either affect growth or elicit a colourful phenotype allowing strains to be selected or screened with relative ease. However, the vast majority of metabolites of interest to industry exhibit no easily selectable or screenable phenotype[2,22]. Instead, mass spectrometry coupled to chromatography is the gold standard for quantifying production of industrially relevant metabolites like these[23], but the time required for each sample is significant, limiting its use for screening large strain libraries that are quickly generated by methods like SCRaMbLE. In this study we aim to overcome this screening bottleneck by employing rapid and semi-automated technologies into a streamlined workflow that is broadly applicable to many metabolic engineering applications, using strain diversification by SCRaMbLE as an ideal test case.

Notably, our previous work has indicated the SCRaMbLE generates improved strains at a rate of approximately one in a hundred[18]. To be able to use SCRaMbLE to optimise yeast for the production of all industrially relevant metabolites, a new approach is needed that enables thousands of samples to be screened quickly by mass spectrometry, in order to accelerate the discovery and investigation of the chromosomal rearrangements that lead to increased performance. Here, we present a full and adaptable workflow designed to work with SCRaMbLE to identify and characterise strain libraries with rapid liquid chromatography coupled to mass spectrometry (LC-MS) quantification. We focus on heterologous production of betulinic acid (BA), a triterpenoid of significant industrial interest only detectable by LC-MS, and demonstrate screening of 1000 strains in only 24 h to identify a dozen top performing candidates with improved BA production.

Core to our workflow is ultra-fast LC-MS, a modified version of a recent technique developed for therapeutic drug monitoring in patient blood plasma[24]. This uses only a guard column in place of the standard liquid chromatography analytical column and allows each sample to be processed sevenfold faster than with standard LC-MS[24]. Following LC-MS screening we utilise multiplex nanopore sequencing to quickly resolve selected strain genotypes and use this information to determine genetic changes that contribute to improved performance. This workflow is designed to complement the speed of SCRaMbLE by quickly isolating top performing strains and characterising these genetically as well as for the product of interest.

## Results

**Ultra-fast LC-MS screening of 1000 strains.** Our post-SCRaMbLE screening workflow consists of semi-automated colony picking and sample preparation, ultra-fast LC-MS measurement, and multiplexed nanopore sequencing. To assess our workflow, we first constructed and verified a heterologous pathway for BA biosynthesis in S. cerevisiae. Biosynthesis of BA at a detectable titre was achieved by introducing three heterologous genes (AtLUS1, AtATR1, and BPLO) and overexpressing ERG9 and a truncated version of HMG1 (tHMG1). This was constructed in a yeast strain containing a synthetic chromosome V (synV), with 176 LoxPsym recombination sites throughout the chromosome[25] (Fig. 1a, left).

AtLUS1, AtATR1, BPLO, and tHMG1 were all expressed under control of strong promoters and ERG9 under control of a medium strength promoter (Fig. 1a, right). The resulting strain, yGG066, also containing synV[25], produced BA to 0.28 µg/gCDW when grown under the conditions used in this study. To trigger SCRaMbLE, yGG066 was transformed with a Cre recombinase expression plasmid (pSCW11-creEBD, LEU$^+$). SCRaMbLE was then induced for 4 h by addition of 1 µM beta-estradiol, before cells were washed and plated. After 3 days of growth on URA$^-$ agar, 70 uninduced control colonies and 964 induced colonies were picked by an automated PIXL robot (Singer Instruments) taking under 15 min per 96-well plate. Colonies were picked into 11-deep well 96-well plates containing 500 µl SDO URA$^-$ media and grown overnight with shaking. This was used to reinoculated 11 plates containing fresh media (1:100 dilution) with the remaining culture stocked at −80 °C in glycerol. The reinoculation was grown for 2 days prior to solvent extraction of BA. An automated protocol for solvent extraction of BA was developed using a CyBio FeliX robot, which reduced the time of extraction to under 30 min per plate and removed variation due to human error.

The gold standard for BA detection has previously been LC-MS[26,27], but as this takes ~5 min per sample, screening 1000 strains would take more than two standard working weeks (~83 h). To tackle 1000 samples we adapted an ultra-fast LC-MS method that only uses a guard column as opposed to a standard chromatography column[24]. The guard column provides sufficient chromatographic separation for measurement with a retention time of only 40 seconds for BA, compared to ~4.8 min for conventional LC-MS. With a runtime of 72 s with 12 s of injection time, the guard column reduces the time for screening from ~5 min per well (~8 h per 96-plate) to 84 s per well (<2.3 h per 96-plate) offering a throughput advantage of ~4-fold (Fig. 1b). This rapid method maintains comparable ion separation, even for complex samples without filtering (Supplementary Fig. 5), and has sufficient assay robustness with minimal sample drift even over 11 plates (Supplementary Fig. 1).

Total cell growth was measured prior to BA extraction to normalise for any growth differences that may be caused by SCRaMbLE. BA titres were calculated using BA standards spiked into negative control biological samples (10 µg/ml – 1 ng/ml) and normalising by cell dry weight (inferred from OD$_{600}$). A threshold for significant titre improvement was set 4 standard deviations above the mean control values (Fig. 1c). A total of 12 screened strains exceeded this threshold. These strains all grew slower than the pre-SCRaMbLE controls (even when cured of the

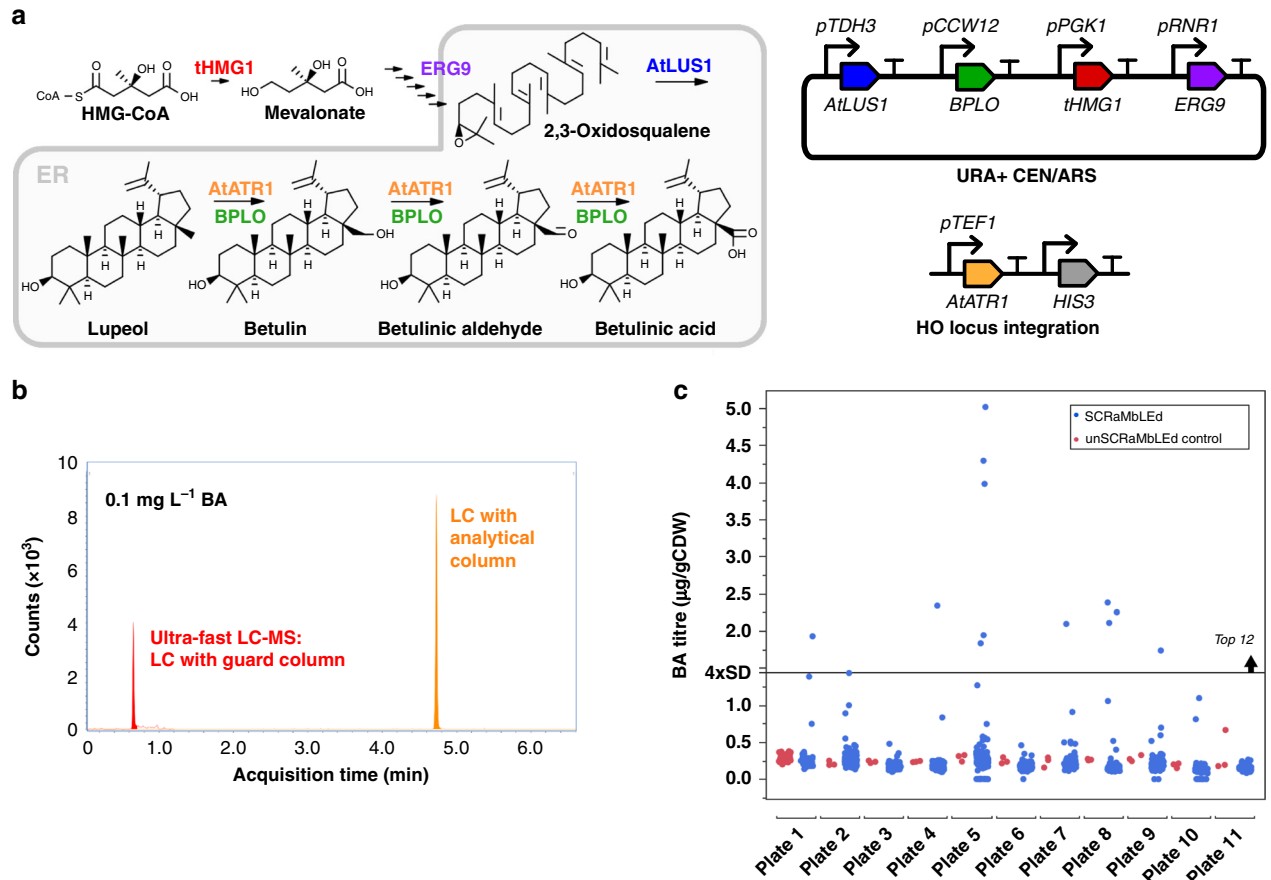

**Fig. 1 SCRaMbLE of a betulinic acid-producing strain followed by rapid screening yields a diverse library. a** BA is synthesised by redirecting flux from the endogenous mevalonate pathway (left, top row) using three heterologous enzymes (AtLUS1, a lupeol synthase from *Arabidopsis thaliana*; BPLO, a cytochrome P450 from *Betula platyphylla*; and AtATR1, a P450 reductase from *Arabidopsis thaliana*). tHMG1 and ERG9 were additionally introduced to increase flux down the mevalonate pathway. Four genes (*AtLUS1, BPLO, tHMG1,* and *ERG9*) were expressed from a URA+ CEN/ARS plasmid while *AtATR1* was integrated into the genome at the HO locus on chromosome IV (right). **b** Ultra-fast LC-MS utilises only a guard column for separation (red) which analyses each sample with a retention time of 40 s compared to ~4.8 min for conventional LC-MS (orange). **c** Eleven plates were run with ultra-fast LC-MS over 26 cumulative hours split across 2.5 days. Forty pre-SCRaMbLE control samples (red) were present in the first plate and three in every subsequent plate. Standard curves were run at the start and the end of the screen. After screening 964 SCRaMbLE colonies a minimum $OD_{600}$ cut off of 0.1 was applied. 914 SCRaMbLE strains exceeded this threshold and were plotted (blue). A threshold of above ×4 control standard deviations was set to identify significantly improved strains. Source data are provided as a Source Data file.

BA-production plasmid) but produced more total BA even when not normalised by OD600 (Supplementary Figs. 2 and 3).

**Multiplexed nanopore sequencing of top SCRaMbLE strains**. The top 12 BA-producing strains (BC01-12) identified by the screen were selected from the glycerol stock plates. All strains were checked for any remaining recombinase plasmid, and in two strains where this was found (BC01 and BC03), the plasmid was removed by curing before further work. The top 12 strains were then re-analysed with ultra-fast LC-MS (n = 7) exactly as before to better characterise their improved BA titre phenotypes. All 12 strains did indeed show significantly higher BA titres of between two- and sevenfold above that of the yGG066 control (Fig. 2a). Despite the use of plasmid based-pathway expression in these strains, BA biosynthesis was still maintained over 6 days of continuous culturing (BC01, 02, 03, 07, and 11 tested), with only 1 in 30 isolates losing BA biosynthesis (Supplementary Fig. 7). Although the auxotrophic selection of the plasmid used in this study requires no addition of chemicals, genomic integration of

the pathway genes would be more desirable for future industrial use.

To determine the genotypic cause of the improved phenotypes it is necessary to map the chromosomal rearrangements of the synthetic chromosome in each strain. Oxford nanopore sequencing yields long reads ideal to bridge chromosomal rearrangements that could be missed or difficult to infer by short read sequencing[18,28]. Therefore, we next used a barcoding approach to multiplex all 12 strains and sequenced these together on a single MinION nanopore flow cell. In total, we achieved 11.5 Gbp of reads which, after barcode deconvolution, resulted in an average of 886 Mbp of read data per strain, representing 73.2-fold coverage of the yeast genome (Fig. 2b). Raw reads were aligned to the reference synV chromosome using a LASTAL algorithm in order to quickly identify SCRaMbLE events (Supplementary Fig. 4). Many strains exhibited complex rearrangements involving combinations of deletions, inversions, and duplications, in some cases with regions over 15 kb in length rearranged (Supplementary Data 4). Notably, no obvious rearrangement was detected by our sequencing approach for three of the strains. For these we

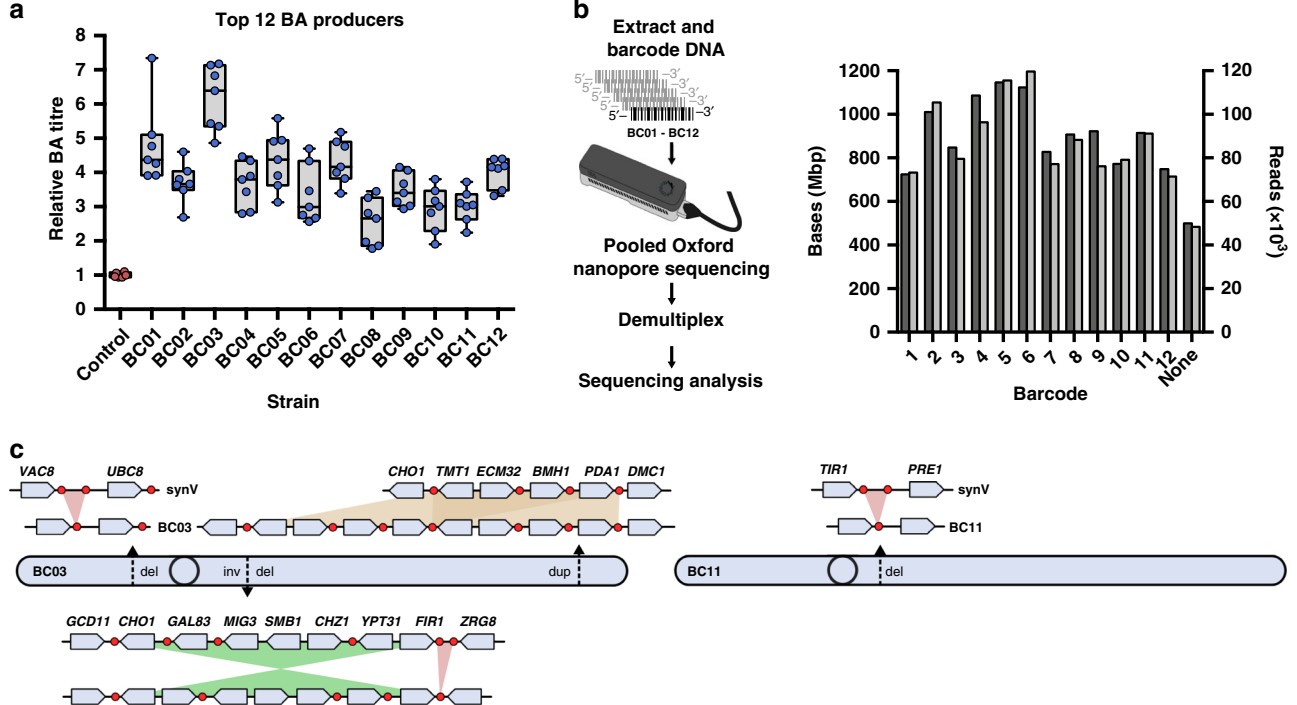

**Fig. 2 Phenotype and genotyping of the top 12 BA producers by ultra-fast LC-MS and multiplexed nanopore sequencing. a** The top 12 strains from the initial screen (BC01–BC12), and the control strain (yGG066), were subject to reanalysis with $n = 7$ ($n = 6$ for the control strain) biologically independent samples using the same extraction and LC-MS workflow as in the initial screen. Data are presented as a box plot centred around the mean with bounds between the 25th and 75th percentile. Whiskers represent minima and maxima. **b** DNA from each of the top 12 strains was extracted and barcoded with 12 unique barcodes (BC01–BC12). The pooled DNA was then sequenced over 48 h. Throughput of bases (dark grey) and reads (light grey) was monitored to ensure no bias was introduced for a particular barcode. **c** Sequence analysis of the highest BA-producer, BC03 (left), indicates four SCRaMbLE events across three loci affecting ~20 kb of the chromosome. In contrast, BC11 (right) exhibits only a single non-coding deletion of 723 bp. Source data are provided as a Source Data file.

speculate that natural mutations elsewhere in the genome may also have yielded beneficial improvements.

Our highest performing strain (BC03) is capable of producing ~7-fold more BA per cell than the pre-SCRaMbLE control strain in 500 μl shaking cultures (Fig. 2a) and ~3-fold more in 50 ml shaking flasks over 68 h (Supplementary Fig. 8). Sequencing of this strain revealed a complex combination of four SCRaMbLE rearrangements on synthetic chromosome V. Two non-coding deletions of 551 and 391 bp, respectively, were seen alongside a 10192 bp inversion and 8724 bp duplication (Fig. 2c, left). In contrast, another strain, BC11, showed just a single small deletion of 723 bp of non-coding DNA, which led to over threefold increased BA titre in 500 μl shaking cultures (Fig. 2c, right).

**A genotype-to-phenotype relationship is established.** To verify a genotypic cause for increased BA titres we sought to artificially recreate a SCRaMbLE event in a control strain. While BC03 was our top producer, our previous efforts have found it extremely challenging to use genome engineering to recreate large inversions and duplications[18]. Such work may also be of limited value when looking to establish a specific genotype-to-phenotype relationship, as it is likely that only a subset of the affected genes is responsible for the improved titre. Instead, for simplicity, we chose to re-create the single small non-coding deletion in BC11 that only removes the 3′ untranslated region of *TIR1* (Fig. 3a; confirmed by PCR in Supplementary Fig. 6).

Quantitative PCR (qPCR) analysis was first used to determine the effect of this deletion on the expression of surrounding genes. At mid-exponential phase *TIR1* exhibits 42% lower expression, while *PRE1* exhibits 67% higher expression compared to the pre-

SCRaMbLE control strain (Fig. 3b). Next, we used CRISPR-based genome engineering to artificially re-create this deletion in the control yGG066 strain, removing the *TIR1* 3′UTR region in synV as seen in the BC11. The re-created deletion was confirmed by PCR (Supplementary Fig. 6) and this strain, 'yGG066 d(*TIR1*-3′)', was subject to LC-MS characterisation exactly as before along with the pre-SCRaMbLE yGG066 strain (negative control) and BC11 (positive control) (Fig. 3c). We observe the same BA titre as in BC11, confirming that this single SCRaMbLE event is responsible for the threefold increase in BA biosynthesis. Revealing this non-intuitive strain improvement was only made possible due to the workflow devised here that links post-SCRaMbLE high throughput LC-MS screening with multiplexed long read sequencing (Fig. 3d).

## Discussion

Rational strain engineering approaches focus on systematically up- or down-regulating endogenous genes, typically informed by mathematical modelling. Inherent in this approach is bias based on prior knowledge of metabolic networks. An alternative, and complementary, approach is to employ whole-genome mutation strategies and other 'black-box' methods that generate unpredictable alterations to the strain genotype[29,30]. SCRaMbLE is one of the most drastic of these, offering a quick and simple way to explore enormous design spaces of the host genotype[30].

Unfortunately for metabolic engineering projects, these methods are only as powerful as the throughput of the screening approach employed to select best performers. The mass spectrometry-based screening workflow developed here offers increased throughput without compromising robustness and accuracy. This workflow

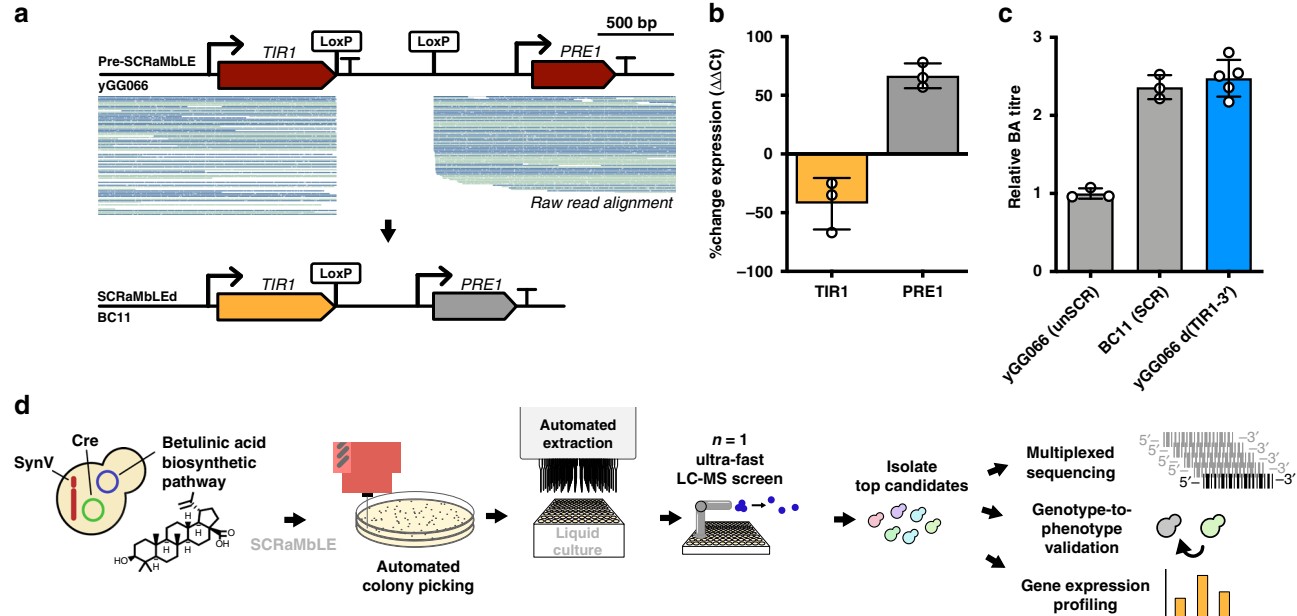

**Fig. 3 Characterisation of BC11 indicates a genotype-to-phenotype relationship. a** SCRaMbLE strain BC11 exhibited a single recombination event between adjacent LoxPsym sites. This was identified by aligning all raw reads to a pre-SCRaMbLE reference sequence (blue and green lines show individual reads, coloured by direction). The inferred SCRaMbLE event was a deletion of the TIR1 3′ UTR (confirmed by PCR, Supplementary Fig. 6). **b** Quantitative PCR was performed on the SCRaMbLE BC11 strain in biological and technical triplicates. The mean of three technical repeats for each biological repeat ($n = 3$) was used to find the biological replicate mean and standard deviation (shown here). **c** BA titre for yGG066 (negative control, $n = 3$ biologically independent samples) and BC11 (positive control, $n = 3$ biologically independent samples) is shown compared to yGG066 d(TIR1-3′) (blue, $n = 5$ biologically independent samples). Mean and standard deviation shown. **d** The entire workflow developed in this study is shown. Following mass genome diversification strains are screened using ultra-fast LC-MS to identify the top candidates. These candidates are then subject to downstream characterisation to gain new insight (multiplexed sequencing, genotype-to-phenotype validation, and gene expression analysis). Source data are provided as a Source Data file.

was designed to accelerate strain characterisation post-SCRaMbLE, where typically only 1% of colonies show improvement. However, it could easily be applied to other black-box methods that diversify strain genotypes, such as the mating-based methods commonly used with budding yeast.

Interestingly, black-box methods like SCRaMbLE often reveal unconventional strategies to improve biosynthesis performance, and these can greatly help inform further rational design. In BC11, we identified that a small non-coding DNA deletion that lowers expression of *TIR1* and increases *PRE1* expression can boost BA titres by threefold, a non-obvious finding. Tir1p is a cell wall mannoprotein and previous studies have shown that lowered expression of *TIR1* is a stress response to both cold- and heat-shock[31]. We suspect that BA accumulates at the cell wall and therefore any changes to the cell wall structure may affect the accumulation of BA. Pre1p is the beta subunit of the 20S proteasome in yeast and is associated with the endoplasmic reticulum[32]. Further larger scale changes were observed in other SCRaMbLE strains leading to increased BA production (Supplementary Data 4). These genes are unlikely to have been identified in a rational search for candidate genes for systematic modification, but the increased BA titre in BC11 persisted when we recreated this deletion in the control strain, demonstrating it to be solely responsible for the phenotype change.

Central to the success of the workflow devised here is the ultra-fast LC-MS guard column method, which allowed us to characterise samples every 84 s, fourfold faster than when using a standard chromatographic column. Unlike other rapid MS techniques such as Rapid-Fire[33] (Agilent), the method presented here requires no additional equipment or infrastructure. Furthermore, unlike the method presented here, other MS techniques such

as REIMS[34] or MALDI-TOF-MS[23,35] lack any chromatographic separation and so may struggle to resolve some metabolites. We anticipate that the ultra-fast LC-MS method used in this study could replace many standard LC-MS workflows seeking to quantify a broad range of metabolites.

The use of automation with this method allowed us to pick, grow, extract, and analyse 1000 strains in 26 h with minimal input from the user, ensuring the screen was robust, rapid, and bias-free. Scaling this to cover larger library sizes would simply require scaling the timescale of this screening or increasing multiplexing through automation capabilities. Crucially, our workflow requires no significant new instrumentation and is therefore likely to be immediately implementable at the increasing number of Bio-foundries found around the world[36].

## Methods

**Strains and media**. *Escherichia coli* Turbo Comp cells (NEB) were used for standard bacterial cloning and plasmid propagation with growth in Luria Bertani media (VWR). Yeast strain yXZX846 (BY4741 with synthetic chromosome V, here referred to as synV) was generated previously[25]. synV yeast was grown in YPD ($10 \, g \, l^{-1}$ yeast extract, $20 \, g \, l^{-1}$ peptone, $20 \, g \, l^{-1}$ glucose) or SDO ($6.7 \, g \, l^{-1}$ yeast nitrogen base, $1.4 \, g \, l^{-1}$ yeast synthetic drop out medium supplement without histidine, leucine, tryptophan and uracil, $20 \, g \, l^{-1}$ glucose) supplemented with histidine ($20 \, mg \, l^{-1}$), leucine ($120 \, mg \, l^{-1}$), tryptophan ($20 \, mg \, l^{-1}$), and uracil ($20 \, mg \, l^{-1}$) as necessary. Two per cent bacteriological agar (VWR) was added to YPD or SDO media when necessary. Unless otherwise stated, all media components were supplied by Sigma Aldrich. SCRaMbLEd yeast strains were stocked in glycerol to a final concentration of 25% (v/v) at −80 °C. $OD_{600}$ measurements were performed using a Synergy HT (BioTek) microplate reader in clear bottom 96-well plates.

**Plasmids and yeast transformation**. Yeast strain yGG037 (synV with AtATR1 integrated at the HO locus with HIS marker) was constructed by transforming synV with plasmid pGG127 (Supplementary Fig. 1, Supplementary Data 2),

linearised by *Not*I digestion (NEB), followed by selection on HIS⁻ SDO agar. yGG066 (BA-producing control strain) was constructed by transforming yGG037 with pGG052 (Supplementary Fig. 1, Supplementary Data 2) followed by selection on URA⁻ SDO agar. pGG127 and pGG052 were constructed using the MoClo Yeast ToolKit (YTK). DNA parts not already in the original YTK collection were synthesised by GeneArt (listed in Supplementary Data 3). pSCW11-*creEBD* was generated in a previous study[13].

**SCRaMbLE**. yGG066-pSCW11-*creEBD* was grown overnight in SDO URA⁻ LEU⁻ media (30 °C, 250 r.p.m. shaking). Culture was diluted to $OD_{600}$ 0.2 in 5 ml SDO URA⁻ LEU⁻ media and grown for 4 h (30 °C, 250 r.p.m. shaking). Beta-estradiol was added to the experimental culture (SCRaMbLE) (final concentration 1 µM) and 1 µl ethanol to the control culture (pre-SCRaMbLE) and both grown for 4 h (30 °C, 250 r.p.m. shaking). Cells were washed twice by centrifugation (3220 r.c.f., 5 min) and resuspension in URA⁻ selective media. Cells were diluted in SDO URA⁻ media ($10^{-4}$, SCRaMbLE culture and $10^{-5}$ pre-SCRaMbLE culture) and spread onto SDO URA⁻ agar media. Plates were left at 30 °C for 3 days.

**Automated colony picking**. Cells were picked from agar dishes into 2 ml 96-deep well plates (Grenier Bio-One) containing 500 µl SDO URA⁻ media using the PIXL automated colony-picking robot (Singer Instruments). 40 pre-SCRaMbLE control strains and 48 SCRaMbLE strains were picked into the first 96-well plate. Subsequent 96-well plates contained three pre-SCRaMbLE control strains each. Wells were reserved for a spiked standard curve on the first plate. Colonies were automatically identified by the PIXL imaging software (version 2.18.0920.1/2.12) prioritising minimum proximity to minimise the risk of picking contamination.

**Automated extraction of BA**. Prior to extraction colonies were picked into SDO URA⁻ media and grown overnight (30 °C, 250 r.p.m.) before being diluted 1:100 into fresh media and grown for 2 days (30 °C, 250 r.p.m.). $OD_{600}$ measurements were taken and negative control wells (synV) were spiked with 10-fold dilutions of a BA standard (Sigma Aldrich) prepared in isopropanol (10 µg/ml–1 ng/ml). Liquid handling steps were developed on CyBio® Composer (Analytik Jena) for execution on a FeliX robot (R 96/250 µl pipetting head, Analytik Jena). Cells grown in a 96-deep well plate (96-DWP) were centrifuged (10 min at 2250 r.c.f.) and the supernatant removed. A 96-DWP was filled with 96 × 1 ml isopropanol (IPA). From this 200 µl IPA was added to the cell pellet using a clean set of 96 tips (CyBio® RoboTipTray 96–250 µl). Cells were vortexed to resuspend in IPA (2000 r.p.m., 2 min, Bioshake QInstruments) and centrifuged (5 min, 2250 r.c.f.). 150 µl supernatant was aspirated and transferred to a 96-well round-bottom plate (Agilent). A further 200 µl IPA was added to the cell pellet and previous resuspension and centrifugation steps repeated. A final volume of 200 µl supernatant was transferred to the same 96-well round-bottom plate. This plate was dried down in a fume hood and resuspended in 70 µl fresh IPA (fivefold concentration) and stored at −20 °C until LC-MS analysis.

**Ultra-fast LC-MS**. Agilent 1290 LC and 6550 Q-ToF mass spectrometers, with electrospray ionisation in negative polarity, were used to detect and measure BA. The MS data acquisition rate was 10 spectra/s. A Zorbax Eclipse Plus C18 UHPLC guard column (2.1 × 5 mm, 1.8 µm particle size; Agilent Technologies, Santa Clara, CA, USA) was used for analysis. LC buffers used were as follows: solvent A, 50% of methanol in 0.1% (v/v) formic acid in water; solvent B, 0.1% (v/v) formic acid in acetonitrile (B). A 0.2 µl injection volume was used for samples and standards. The LC gradient was 72 s in total without post-time run as summarised in Supplementary Data 1. Quantification by LC-MS was based on the peak area of accurately measured deprotonated BA, [M-H]⁻, extracted with a window of ±50 ppm.

**CRISPR knockout**. To generate the TIR1-3′ knockout strain (yGG066 d(*TIR1*-3′)) a B*pi*I-digested cas9 plasmid (pWS2082, LEU⁺, Supplementary Data 2) was transformed into yGG066 along with B*pi*I-digested gRNA plasmid gGG002. gGG002 was generated by phosphorylating (standard T4 PNK (NEB) reaction) and annealing primers GG128 (AGATTTAGGTGCACAAAATGAACA) and GG129 (AAACTGTTCATTTTGTGCACCTAA) followed by a BsmBI golden gate reaction (with MoClo YTK protocol[5]) with pWS2069 (Supplementary Data 2). Donor DNA was generated by PCR amplification of the *dTIR1*–3′ region of BC11 using primers GG126 (GTATTCCTCACGATTAGAACCAGCC) and GG127 (CGTAAAGATTT CGCTGCGAAAG).

**Nucleic acid isolation**. DNA was isolated from BC01-12 using QIAGEN genomic tips 100/G for Oxford Nanopore Sequencing using wide-bore tips. During cloning all plasmids were isolated from bacteria using QIAGEN QIAprep Miniprep kits. For qPCR, RNA was isolated using RiboPure Yeast RNA purification kit (Thermo Scientific) from 50 ml shaking cultures in the mid-log phase (250 ml baffled flasks, 30 °C, 180 r.p.m.).

**Nanopore sequencing**. Extracted DNA from BC01-12 was prepared for Oxford Nanopore sequencing using genomic DNA kit SQK-LSK109 with 1D Native Barcoding kit EXP-NDB104 (Oxford Nanopore Technologies). Prepared DNA was sequenced using an R9.4.1 flowcell in a MinION Mk1B device. Sequencing was run for 48 h with local basecalling in MinKNOW. Sequencing reads relating to Fig. 2b, c, and Fig. 3a are publicly available at the Sequence Read Archive (SRR9988267).

**Sequence analysis**. Reads were demultiplexed using Porechop (https://github.com/rrwick/Porechop) and aligned to pre-SCRaMbLE synV sequence using LAS-TAL (-l100 flag) (http://last.cbrc.jp/). Output files were converted to indexed BAM files using maf-convert (http://last.cbrc.jp/) and samtools (http://samtools.sourceforge.net/). Alignments were searched for SCRaMbLE events using Tablet[37] using a synV.gff file to identify recombination events aligning with LoxPsym boundaries.

**Quantitative PCR**. qPCR was performed in a MasterCycler ep RealPlex 4 (Eppendorf) using a SYBR Green JumpStart qPCR kit. High Capacity cDNA RT kit (Thermo Scientific) was used to generate cDNA. Primers were designed using the IDT RealTime PCR Design Tool (http://idtdna.com/scitools/applications/realtimepcr). PRE1: GG116 (CTTCTAAGGCAGTCACAAGAGG), GG117 (TGGCTTGAATGTACTCGGC); TIR1: GG118 (AGCTGGTGTTTTGGATA TCGG), GG119 (ACCAGCAAAGTCAACCTCAG); and housekeeping gene ACT1: GG114 (TCGAACAAGAAATGCAAACCG), GG115 (GGCAGATTCCAAACCCAAAAC).

**Reporting summary**. Further information on research design is available in the Nature Research Reporting Summary linked to this article.

## Data availability
The source data underlying Figs. 1c, 2a, b, and 3b–c and all supplementary figures are provided as a Source Data file. All Oxford Nanopore raw reads are deposited in the Sequence Read Archive database (accession: SRR9988267). Any other relevant data is available from the authors upon reasonable request.

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

## Acknowledgements

We thank Christopher Benton (Agilent) for assistance in setting up ultra-fast LC-MS capabilities at SynbiCITE. We also thank Dr. Rochelle Aw for assistance in planning and running the qPCR experiment. We thank Prof. John Dueber for distribution of the MoClo YTK kit and Dr. William Shaw for construction of pre-assembled YTK vectors. This work was funded in the UK by BBSRC awards BB/P504579/1, BB/L027852/1, and BB/R002614/1, and EPSRC awards EP/S001859/1 and EP/L011573/1.

## Author contributions

G.-O.F.G. conceived and designed the experiments. G.-O.F.G., L.S., D.B. and S.M.C. designed and performed sample preparation for, and execution of, LC-MS analysis. LC-MS data analysis was performed by G.-O.F.G., S.M.C. and D.B. G.-O.F.G. constructed strains and performed nanopore sequencing with subsequent analysis. M.K., D.T., T.E. and D.M. assisted in interpreting results. G.-O.F.G. and T.E. prepared the manuscript, while all other authors contributed to manuscript improvement.

## Competing interests

GSK support G.-O.F.G. through a BBSRC industrial CASE studentship. Singer Instruments provided SynbiCITE with access to a PIXL robot for a limited time. No industrial contributors had input into initial study design nor on decision of when and where to publish this manuscript. All other authors declare no competing interests.
