## [Peer Review File · Nature Communications]

Reviewers' Comments:

Reviewer #1:

Remarks to the Author:

In the manuscript "Improved betulinic acid biosynthesis using synthetic yeast chromosomes, multiplexed nanopore sequencing, and an ultra-fast LC-MS method", the authors describe a rapid screening workflow to combine automated sample preparation, a new ultra-fast 40-second LC-MS method, and barcoded nanopore sequencing to isolate and describe the genetic changes of the best performing strains. They demonstrated that production of the triterpenoid betulinic acid could be optimized by inducing SCRaMbLE in strains containing synV. Among 1000 clones, betulinic acid titre is improved in 12 strains, between 2- to 7-fold. In one of the strains, the authors found a single non-coding deletion of TIR3 3' UTR could lead the desired phenotype.

The most broadly interesting aspect of this manuscript to me is the semi-automatic screen pipeline, which could be applied to identify strains not only producing betulinic acid but also many other natural products. It greatly reduces the time and effort for metabolic engineering. The SCRaMbLE of synV, although quite interesting, especially the identification of TIR1 in regulating betulinic acid biosynthesis, has been demonstrated in their early paper.

In general, I think the manuscript is a well-written and could be a nice plus of the Sc2.0 collection. I have several concerns:

1. Since demonstration of the SCRaMbLE system has been well-studies in the last batch of Sc2.0 collection, I would expect to see more mechanistic insights of the identified strains. How TIR1 is involved in betulinic acid biosynthesis pathway should be addressed.
2. Besides TIR1, what happened in the other 11 strains? What leads to the improvement of betulinic acid biosynthesis?
3. The improvement of betulinic acid in the identified strains is still quite limited. How stable these clones are? The authors mentioned that the strains grow slower. Is it caused by SCRaMbLEd synV or metabolic burden?

Minor points:

1. Line 69, 100e18? Or 10e18?

Reviewer #2:

Remarks to the Author:

Overview

This manuscript considers a rapid screening workflow for SCRaMbLE to increase production phenotypes. The key advance here is to combine a new ultrafast LC-MS method to isolate the best performing strains. which is likely useful for developing High-throughput screening methods in synthetic biology.

While the paper contains some interesting elements, there are quite a few problems with the experimental setup and manuscript discussion that need to be resolved.

This paper needs to include many more details to assess the impact.

Major

Comment 1. The topic is not clearly to emphasize the key point of the work;

Comment 2. I noticed that you assembled AtLUS1, BPLO, tHMGE and ERG9 in a single copy plasmid while integrate the AtATR1 in the HO locus. In general, it is possible to loss plasmids during fermentation process, which may affect the titer of desired products(Tyo et al. Nat Biotechnol. 2009). Add fermentation data of the BC01-12 in shaking flask.

Comment 3. In Jin et al. Microb Cell Fact. 2019, it looks like that this betulinic acid pathway can generate some by-products, such as the betulinic aldehyde and the betulin. Could the betulinic acid be separated from these by-products by the guard column ? Add mass spectra and total ion current maps in supplementary materials to determine the validity and accuracy of the data.

Minor

Comment 1. It is crucial to extract target product with high quality. How to avoid mass spectrometry pollution without filtering samples?

Comment 2. Beyond RT-PCR, it would be helpful to see transcriptomics to assess the impact of the modifications.

Comment 3. Give the detail position of the SCRaMbLE event in the strains of Supplementary Figure 5.

Comment 4. Discuss the potential reasons of BA titer improvement in these non-SCRaMbLE strains (BC05, 06, 12).

Reviewer #3:

Remarks to the Author:

I was asked specifically to comment on the ultra-rapid LC-MS aspect of this study.

There is mention throughout the paper of '40 seconds' as the rate of analysis, yet Supplementary Table 1 and Line 457 suggest a total analysis time of 1.2 min (72 s). Was a dual-needle sampler used for this application? This makes quite some difference to the inject-to-inject cycle time, and if this was implemented, should be included in the manuscript to aid others reproduce the work. Is 40 seconds the retention time of the peak of interest?

There is a minor error in Supplementary Table 1 (cell C4 states 5 %, I assume this should read 2 %).

I have only one chromatogram to review, but in my experience the peak shape/efficiency looks comparable to what we were able to achieve in our previous work. The TOF settings (10 Hz) are appropriate given the efficiency of the separation.

It is excellent to see further application of ultra-rapid LC-MS. I think this will be a valuable addition to the literature in this regard.

Reviewer #1 (Remarks to the Author):

In the manuscript “Improved betulinic acid biosynthesis using synthetic yeast chromosomes, multiplexed nanopore sequencing, and an ultra-fast LC-MS method”, the authors describe a rapid screening workflow to combine automated sample preparation, a new ultra-fast 40-second LC-MS method, and barcoded nanopore sequencing to isolate and describe the genetic changes of the best performing strains. They demonstrated that production of the triterpenoid betulinic acid could be optimized by inducing SCRaMbLE in strains containing synV. Among 1000 clones, betulinic acid titre is improved in 12 strains, between 2- to 7-fold. In one of the strains, the authors found a single non-coding deletion of TIR3 3’ UTR could lead the desired phenotype.

The most broadly interesting aspect of this manuscript to me is the semi-automatic screen pipeline, which could be applied to identify strains not only producing betulinic acid but also many other natural products. It greatly reduces the time and effort for metabolic engineering. The SCRaMbLE of synV, although quite interesting, especially the identification of TIR1 in regulating betulinic acid biosynthesis, has been demonstrated in their early paper.

We thank the reviewer for the positive assessment of this manuscript and experimental work. We entirely agree that the semi-automatic screen pipeline is the most broadly interesting aspect and we anticipate this to be of interest to the broad readership of Nature Communications. We sympathise that SCRaMbLE of synV has been addressed in our previous paper, hence our focus on the semi-automatic workflow with SCRaMbLE of synV as a test-case. However, we believe this manuscript also represents a first and significant move away from ‘easily screenable’ phenotypes (GFP, carbon sources, violacein, carotene etc.). This, leveraging the semi-automatic screening workflow presented here, is the main focus of this manuscript, rather than simply the SCRaMbLE of synV.

In general, I think the manuscript is a well-written and could be a nice plus of the Sc2.0 collection. I have several concerns:

1. Since demonstration of the SCRaMbLE system has been well-studied in the last batch of Sc2.0 collection, I would expect to see more mechanistic insights of the identified strains. How TIR1 is involved in betulinic acid biosynthesis pathway should be addressed.

We thank the reviewer for their comments. While we do agree that mechanistic insight of SCRaMbLE events linking to phenotype would be of great interest, we believe this work is well beyond the realm of this manuscript and would additionally distract from the main focus: which is the semi-automatic screening workflow we describe. We share the enthusiasm of the reviewer for seeing this experimental work done as no SCRaMbLE paper has yet to properly address this question. For this reason, we plan to properly answer this question in a separate forthcoming manuscript that establishes an ‘omics’ workflow to interpret phenotypes resulting from SCRaMbLE genomic changes.

For now, however, in addition to the discussion section speculating on the contribution of TIR1 and PRE1 to BA production, we have also added new text to the discussion to further speculate on the role of TIR1 in BA production (lines 267-269). Additionally, we have expanded **Supplementary Table 4** to now list the affected genes in each SCRaMbLE event and an additional column titled “*speculation on hypothetical genotype-phenotype relationships*”. Here, we provide speculation on how suspected changes to certain genes, as a result of SCRaMbLE, might affect the BA titre. While this is certainly not the focus of the manuscript, we hope this provides the reader with valuable food for thought, particularly adding clarity to the fact that SCRaMbLE can yield unpredictable targets for future metabolic engineering efforts. We have referenced changes to **Supplementary Table 4** in the main text (lines 270-271 and 179).

2. Besides TIR1, what happened in the other 11 strains? What leads to the improvement of betulinic acid biosynthesis?

We thank the reviewer for this suggestion. We have updated and relocated (what was) **Supplementary Figure 5** to now include this information. This is now **Supplementary Table 4**. We have also referenced this change in the main text (lines 270-271 and 179). As mentioned above, this table now includes a column to provide speculation on the genotype-phenotype link for the various SCRaMbLE events.

3. The improvement of betulinic acid in the identified strains is still quite limited. How stable these clones are? The authors mentioned that the strains grow slower. Is it caused by SCRaMbLEd synV or metabolic burden?

We thank the reviewer for their comments. To address whether the strains grow slower as a result of the SCRaMbLEd chromosome or the metabolic burden of increased BA production we have cured BC01-12 strains of all plasmids (thus producing no BA) and have performed a growth assay comparing it to a pre-SCRaMbLE synV strain, measuring both the maximum growth rate during a time course and the final maximum OD600 reached. This information has been included in an **updated Supplementary Figure 3**.

We see that when strains are cured of their BA-producing plasmid, the max OD600 at the end of the time course recovers, but the max growth constant during the time course remains low. We therefore conclude that there is metabolic burden due to the plasmid-based pathway as a significantly higher max OD600 is achieved in all cases when the plasmid is removed. However, the data also reveal that the SCRaMbLE of SynV leads to cells that now grow with reduced maximum growth rate, presumably due to the genomic changes. All strains grow significantly slower than the control, even when not expressing the BA-production pathway (text added to line 152).

We have also performed an additional experiment to specifically address the reviewer’s comment about stability. We cultured a 96-well plate with 500 µl of 6 isolates of the SCRaMbLEd cultures (BC01, 02, 03, 07, and 11) and yGG066, (a pre-SCRaMbLE control) and measured BA titre after one 48 hr growth cycle and after 3x 48 hr growth cycles (148 hr). After 6 days of culturing the majority of isolates retain higher BA production than the pre-SCRaMbLE control. For each strain tested

between 1-2 isolates exhibit BA titres that return to the same level of the pre-SCRaMbLE control. Only a single isolate lost the ability to produce BA, which may be due to plasmid loss or mutation. These data indicate that these clones are relatively stable in the culturing conditions used. A comparison of the BA titre between 48 and 148 hrs is shown in Supplementary Figure 7 and now referenced in the main text (lines 162-166).

Minor points:

1. Line 69, 100e18? Or 10e18?

We can clarify that the sentence should read 1 in 100 (citation #18). We appreciate the citation superscript '18' is misleading when appearing after a number. Therefore, we have changed the text to use the word 'hundred' in this instance (line 72).

Reviewer #2 (Remarks to the Author):

Overview

This manuscript considers a rapid screening workflow for SCRaMbLE to increase production phenotypes. The key advance here is to combine a new ultrafast LC-MS method to isolate the best performing strains. which is likely useful for developing High-throughput screening methods in synthetic biology.

While the paper contains some interesting elements, there are quite a few problems with the experimental setup and manuscript discussion that need to be resolved. This paper needs to include many more details to assess the impact.

We would like to thank Reviewer #2 for taking the time to assess this paper and provide constructive comments. We share the reviewer's general enthusiasm for our screening workflow leveraging ultra-fast LCMS which we agree is useful for high throughput screening in synthetic biology.

Major

Comment 1. The topic is not clearly to emphasize the key point of the work;

Unfortunately, we are slightly uncertain as to what this comment is referring to. We have interpreted this comment to mean "*the key point of the work is not clearly emphasised*". Assuming this interpretation, we apologise that the key point is not clear enough. To address this we have reworded, and added additional text, to the abstract and introduction (lines 30-32, 66-69). While this study is naturally multi-faceted we believe the emphasis is on the methodology showcased here which, as the reviewer says, is useful for developing high-throughput screening methods in synthetic biology, with SCRaMbLE chosen as an ideal use-case.

Comment 2. I noticed that you assembled AtLUS1, BPLO, tHMGE and ERG9 in a single copy plasmid while integrate the AtATR1 in the HO locus. In general, it is possible to loss plasmids during fermentation process, which may affect the titer of

desired products(Tyo *et al. Nat Biotechnol. 2009*). Add fermentation data of the BC01-12 in shaking flask.

We thank the reviewer for their insight regarding plasmid loss. While this is possible, during the growth phases prior to BA extraction we were always maintaining the selection pressure for the plasmid. It should be noted that the auxotrophic selection in this study does not require the addition of chemicals (lines 164-166). While Tyo *et al.* nicely discuss the genetic instability of plasmids in *E. coli* strains, we understand that yeast plasmids based on the CEN/ARS replication origin (as in this study) are more stable due to being very low copy, thus mitigating the segregationally instability seen with 2 micron plasmids (Zhang *et al* 1997, [https://doi.org/10.1016/S0734-9750\(96\)00033-X](https://doi.org/10.1016/S0734-9750(96)00033-X)). The CEN/ARS plasmid was chosen for the BA pathway specifically for this reason.

We have added shaking flask data for yGG066 (pre-SCRaMbLE control) and BC03 (top performer of this study) to show that BA production is maintained during a 72 hr fermentation at the shaking flask scale (supplementary figure 8, line 185-186). Additionally we investigated plasmid loss by back-culturing six isolates of six strains (yGG066, BC01, 02, 03, 07, 11) over 6 days with 3x 48hr growth cycles. Over this time only a single isolate of the 30 exhibited plasmid loss (supplementary figure 7 and lines 162-164). This, in combination with our flask data, suggests that over the timescales of this study we observe good plasmid stability.

Comment 3. In Jin et al. Microb Cell Fact. 2019, it looks like that this betulinic acid pathway can generate some by-products, such as the betulinic aldehyde and the betulin. Could the betulinic acid be separated from these by-products by the guard column ? Add mass spectra and total ion current maps in supplementary materials to determine the validity and accuracy of the data.

We appreciate the reviewer's comments regarding the separation of betulinic acid from by-products using this ultra-fast LC-MS technique. MS data was searched by expected m/z for the pathway intermediates mentioned in Jin *et al* 2019. Although no response was detected, it is unlikely that these materials would be ionised under the conditions used, namely negative ion detection electrospray. A potential hydrolysis product was detected, with a retention time of 0.52 minutes and resolved from the betulinic acid, demonstrating the resolution of similar compounds with differing polarity. In our experience, data acquired using the guard column is extremely similar to resolution achieved using 50 mm columns thus giving us further confidence in the ability of the guard column to separate by-products.

We thank the reviewer for their suggestion to include mass spectra and ion current maps to the supplementary material. We have now provided total ion chromatograms, extracted ion chromatograms of target material and a representative mass spectrum for a negative control, a spike negative control, and a low and high expression BA culture (Supplementary Figure 5) and referenced this in the text (line 127-128).

Minor

Comment 1. It is crucial to extract target product with high quality. How to avoid mass spectrometry pollution without filtering samples?

We thank the reviewer for their well-placed concern regarding sample quality. The electrospray LC-MS interface used is tolerant of complex samples, thus we have found that more than eleven 96-well plates in a row can be analysed using the rapid LC approach with no significant loss of sensitivity or contamination. We have updated the main text to reflect this fact (line 127-128).

Comment 2. Beyond RT-PCR, it would be helpful to see transcriptomics to assess the impact of the modifications.

We thank the reviewer for this recommendation and we entirely agree that a full transcriptomics study of SCRaMbLE strains would be ideal in order to fully assess the impact of modifications on phenotypes. However, we believe this substantial further work is well beyond the scope of this particular paper. Indeed, a separate study specifically addressing this question is the focus of a planned manuscript from our groups and others. For the purposes of this current manuscript – where the main focus is the semi-automated screening workflow – we believe RT-PCR to be sufficient.

Comment 3. Give the detail position of the SCRaMbLE event in the strains of Supplementary Figure 5.

We thank the reviewer for this suggestion, we agree that inclusion of this data would improve this manuscript. We have now updated (what was) Supplementary Figure 5 to now include this information as **Supplementary Table 4** and referenced in the main text (lines 269-270 and 179).

Comment 4. Discuss the potential reasons of BA titer improvement in these non-SCRaMbLE strains (BC05, 06, 12).

We thank the reviewer for this suggest. We have updated the text to reference the BA titre in these strains where no obvious SCRaMbLE event was detected (line 180-182).

Reviewer #3 (Remarks to the Author):

I was asked specifically to comment on the ultra-rapid LC-MS aspect of this study.

There is mention throughout the paper of '40 seconds' as the rate of analysis, yet Supplementary Table 1 and Line 457 suggest a total analysis time of 1.2 min (72 s). Was a dual-needle sampler used for this application? This makes quite some difference to the inject-to-inject cycle time, and if this was implemented, should be included in the manuscript to aid others reproduce the work. Is 40 seconds the retention time of the peak of interest?

We thank the reviewer for highlighting this clarification to us. The 40s referred to in the study was indeed referring to the retention time of the peak of interest. We have updated the main text where appropriate to clarify that while the retention time is 40s the total analysis time is 72 seconds with 12 seconds required for injection. This total cycle time of 84s determines the throughput more than the 40s retention time. We apologise for this lack of clarity and have gladly corrected the manuscript accordingly. [lines 27, 80, 122-126, 139-140, 277, and 287]

There is a minor error in Supplementary Table 1 (cell C4 states 5 %, I assume this should read 2 %).

We apologise for this typing error which has gladly been corrected (Supplementary Table 1, cell C4).

I have only one chromatogram to review, but in my experience the peak shape/efficiency looks comparable to what we were able to achieve in our previous work. The TOF settings (10 Hz) are appropriate given the efficiency of the separation.

It is excellent to see further application of ultra-rapid LC-MS. I think this will be a valuable addition to the literature in this regard.

We thank the reviewer for taking the time to specifically assess the ultra-rapid LC-MS aspect of this study. We entirely agree that it is a method that is a valuable addition to the literature and while we focus on a synthetic biology workflow here, we anticipate this MS method being adopted by a broad range of disciplines to improve an equally wide range of MS-based workflows.

Reviewers' Comments:

Reviewer #1:

Remarks to the Author:

All my comments have been addressed.

Reviewer #2:

Remarks to the Author:

The authors have answered all my questions.

Reviewers' Comments:

Reviewer #1:

Remarks to the Author:

All my comments have been addressed.

Reviewer #2:

Remarks to the Author:

The authors have answered all my questions.

We thank the reviewers for the continued enthusiasm and support for the work presented here.

Their useful and constructive comments throughout have ensured this manuscript is of an appreciable higher standard as a result.